# Gastritis Cystica Profunda: A Rare Disease, a Challenging Diagnosis, and an Uncertain Malignant Potential: A Case Report and Review of the Literature

**DOI:** 10.3390/medicina59101770

**Published:** 2023-10-04

**Authors:** Francesca De Stefano, Giorgio M. P. Graziano, Jacopo Viganò, Aurelio Mauro, Andrea Peloso, Jacopo Peverada, Raffaele Fellegara, Alessandro Vanoli, Giuseppe G. Faillace, Luca Ansaloni

**Affiliations:** 1Department of Surgery, Ospedale Citta’ di Sesto San Giovanni, 20099 Sesto San Giovanni, Italy; francesca.destefano1292@gmail.com (F.D.S.); giuseppe.faillace@asst-nordmilano.it (G.G.F.); 2University of Pavia, 27100 Pavia, Italy; giorgiomariapa.graziano01@universitadipavia.it (G.M.P.G.); jacopo.peverada01@universitadipavia.it (J.P.); raffaele.fellegara01@universitadipavia.it (R.F.); l.ansaloni@smatteo.pv.it (L.A.); 3Department of General Surgery I, Fondazione IRCCS Policlinico San Matteo, 27100 Pavia, Italy; j.vigano@smatteo.pv.it; 4Gastroenterology and Endoscopy Unit, Fondazione IRCCS Policlinico San Matteo, 27100 Pavia, Italy; 5Visceral Surgery Division, Organ Transplantation Division, University Hospitals of Geneva, Department of Surgery, University of Geneva, 1205 Geneva, Switzerland; andrea.peloso@unige.ch; 6Anatomic Pathology Unit, Department of Molecular Medicine, University of Pavia, IRCCS San Matteo Hospital Foundation, 27100 Pavia, Italy; a.vanoli@smatteo.pv.it

**Keywords:** gastritis cystica profunda, endoscopic ultrasound, ectopic gastric glands, gastric submucosal lesion, gastric surgery

## Abstract

Gastritis cystica profunda (GCP) has been defined as a rare submucosal benign gastric lesion with cystic gland growth. Due to its unclear etiopathogenesis, this lesion is often misdiagnosed and mistaken for other gastric masses. Currently, a standardized treatment for GCP lesions is still missing. Here, we illustrate a case of a patient admitted to our general surgery department for melena and general discomfort. No history of peptic ulcer or gastric surgery was present. Upper GI endoscopy was performed, showing a distal gastric lesion with a small ulceration on the top. CT-scan and endoscopic ultrasound confirmed the presence of the lesion, compatible with a gastric stromal tumor, without showing any eventual metastasis. Surgical gastric resection was performed. Histological findings were diagnostic for GCP, with cistically ectasic submucosal glands, chronic inflammation, eosinophilic infiltration and foveal hyperplasia. GCP is a very exceptional cause of upper-GI bleeding with specific histological features. Its diagnosis as well as its therapy are challenging, resulting in several pitfalls. Even though it is a rare entity, GCP should always be considered in the differential diagnosis of gastric submucosal lesions.

## 1. Introduction

In 1947, Scott and Payne reported a case of a tumor-like gastric lesion presenting as diffuse cystic hyperplasia in the submucosa of the stomach, which they were the first to describe as Gastritis Cystica Profunda (GCP) [1]. A few decades later, gastritis cystica polyposa was described by Littler and Gleibermann as an uncommon histological finding characterized by many cystically dilated gastric glands in the submucosa and foveolar hyperplasia [2]. They reported this condition as occurring at the gastroenterostomy site, potentially being a reaction of the submucosa of the stomach to gastric surgery.

The definition of GCP proposed by Scott and Payne was then reintroduced by Franzin and Novelli in 1982, due to histological features being similar to a condition of the colon—colitis cystica profunda [3].

GCP is a rare condition and only a limited number of case reports have been published on the topic, mainly regarding the Eastern population. Different hypotheses have been made on the etiology of this lesion, but at present it has yet to be fully clarified. GCP is more commonly found in patients with previous gastric surgery or endoscopic procedures such as polypectomies or biopsies [4,5]. Hence, its etiology may be related to mucosal disruption caused by surgery or the suture materials used during gastroenterostomy, as well as post-surgical chronic inflammation and ischemia. However, more recently, cases of GCP have been reported in patients without a history of gastric surgery [6], suggesting another not-yet elucidated pathogenetic mechanism.

The clinical presentation of GCP is vague, and in most cases, it is an incidental finding [7]. More commonly occurring symptoms are gastrointestinal bleeding, abdominal pain, weight loss, anorexia and gastric outlet obstruction [8,9]. Due to its rare occurrence and its non-specific presentation, diagnosis can be challenging. At present, the pathological examination of the specimen is the gold standard for the diagnosis of GCP. At histology, cystic dilation and hyperplasia of gastric glands occur within the submucosa of the stomach [10].

Many authors [11,12] consider GCP as a precancerous lesion or paracancerous lesion because it is detected together with gastric adenocarcinoma, high grade dysplasia and carcinoma with lymphoid stroma [9,13]. However, no clear pathogenetic mechanisms or robust evidence support this theory.

In this paper, we report a case of a patient presenting with upper gastrointestinal bleeding in the absence of previous gastric surgery. The clinical presentation and the diagnostic workup are analyzed, as well as the therapeutic choice, that led to the histological definitive diagnosis of GCP. Due to its rare occurrence and the lack of robust data, especially in the Western population, we did not take into consideration GCP among the differential diagnoses and a gastric stromal tumor was hypothesized. The pathological examination of the surgical specimen was paramount for the diagnosis. To our knowledge, this is one of the limited cases of GCP occurring in unoperated stomachs in the Western population, and more data are needed in order to standardize the diagnostic and therapeutic process.

## 2. Case Report

A 61-year-old male was admitted to the General Surgery Department due to general discomfort and multiple episodes of melena in the previous 24 h. His medical history was positive for arterial hypertension undergoing therapy with Amlodipine 10 mg/daily, previous major trauma with multiple bone fractures and diaphragmatic laceration treated with explorative laparotomy and suturing of the diaphragm, as well as deep venous thrombosis undergoing anticoagulant oral therapy with a dose of Warfarin adjusted according to INR values. No previous gastrointestinal disorders were reported. Physical examination was unspecific, revealing a diffusely tender abdomen without palpable masses or signs of peritonism. The blood tests at arrival revealed anemia with a hemoglobin value of 8.8 g/dL, a hematocrit value of 27.2% and an international normalized ratio (INR) increased to 3.00, due to the patient’s therapy. The anticoagulant oral therapy was replaced with low molecular weight heparin (LMWH) and a blood transfusion was administered. After a hemotransfusion of two units of packed red blood cells, the patient’s hemoglobin levels increased up to 10.5 g/dL the day after admission.

The patient underwent a diagnostic upper GI endoscopy for melena. The endoscopic examination revealed a 25 mm submucosal lesion at the posterior wall of the antrum (Figure 1), with a superficial ulcerative area 4 mm in diameter, in the absence of active bleeding (Figure 2).

An upper GI endoscopy also revealed a thickening of the folds in the gastric body. Biopsies of the submucosal lesion were performed, and the histological examination showed foveolar hyperplasia without evidence of Helicobacter Pylori infection.

A 50 × 20 mm irregular lesion of the gastric antrum without evidence of metastasis was confirmed with contrast-enhanced computed tomography (CE-CT; Figure 3).

To better characterize the lesion and its extension through the gastric wall, an upper endoscopic ultrasound (EUS) with radial echoendoscope was performed. The EUS showed a 5.0 × 2.8 cm submucosal antral mass with a heterogeneous echo, penetrating the muscolaris propriae with no involvement of the subserosa (Figure 4).

On contrast-enhanced EUS images, after Sonovue^TM^ administration, the septa of the lesion were hyperechoic while the holes remained anechoic (Figure 5).

To further define the nature of the lesion, an EUS with a linear echoendoscope and a fine needle biopsy (FNB) with a 22-Gauge needle were performed. However, the FNB results were inconclusive for a definitive diagnosis. Despite the undiagnostic FNB, the endoscopic, radiological and EUS findings were indicative of a gastrointestinal stromal tumor (GIST). After a multidisciplinary discussion including surgeons, oncologists, endoscopists and radiologists, resection of the lesion was indicated to obtain a definitive diagnosis and to prevent further gastric bleeding in the patient, which would result in long-term anticoagulant therapy. Due to the location and dimension of the lesion, endoscopic resection was not performed, and the patient underwent distal gastric resection with Billroth II-type reconstruction. The surgery had no short-term complications, and the patient was discharged on the eight post-operative day. On the 18th postoperative day, the patient was readmitted to the Surgical Department for sudden onset of abdominal pain and vomiting. The clinical presentation was indicative of bowel obstruction and the CT scan confirmed an incarcerated loop of the proximal ileum secondary to post-operative adhesions. A relaparotomy was performed with adhesiolysis and there was no need for bowel resection due to the vitality of the ileal loop. The residual postoperative period was uneventful, and the patient was discharged after one week.

Macroscopic pathologic examination of the surgical specimen showed a multiloculated cystic lesion of 2 cm covered by normal mucosa, without serous membrane invasion, at the gastric antrum. Microscopic histological findings presented cystically dilated glands, diffuse in the submucosa (Figure 6a–c) and focal in the muscolaris propriae, with foveolar hyperplasia, chronic inflammation and eosinophilic infiltration of the mucosa (Figure 7).

These features were consistent with a diagnosis of GCP. The lymph-nodes examined were microscopically free of neoplastic cells.

The multidisciplinary team suggested no adjuvant treatment and the patient was examined at 18-months follow-up, testing negative.

## 3. Discussion

Gastritis cystica profunda is a rare pathological lesion of the stomach characterized by elongation and hyperplasia of the gastric foveolae, disruption of the muscolaris mucosae and deep epithelial migration and proliferation, with many cystically dilatated glands in the submucosa. In the literature, several terms have been proposed to describe this entity, such as gastritis cystica polyposa, multiple polypoid cystic gastritis, heterotopic cystic malformation of the stomach and diffuse submucosal cysts of the stomach [2,3,7]. GCP is more commonly found in the fundus of the stomach, but it can occur in other gastric regions as well. In the largest case series reported to date, by Xu et al. of 34 patients, the most common location was the cardia, followed by the antrum [14]. In contrast, Congyang Li et al. presented seven cases of GCP that mainly occurred in the fundus, with only one case in the antrum [15]. In our case report, the lesion was in the posterior wall of the antrum, a less common but still possible location for GCP.

The pathophysiology of GCP is yet to be totally clarified. In the literature, prior gastric surgery is described in 65% of cases. In 1981, Franzin and Novelli pointed out that dilated cystic glands in the submucosa were consequent to the post-surgical disruption of the gastric mucosa [3]. Ischemia and chronic inflammation induced by foreign suturing materials represent possible triggers for the disruption of the muscolaris mucosae, epithelial migration and cystic gland dilatation in previously operated-on stomachs, several years after surgery [8,16,17]. Bile reflux during gastrojejunostomy may lead to atrophic gastritis and intestinal metaplasia, which are histopathological features of GCP [18]. Nevertheless, many authors have reported cases of GCP in unoperated stomachs, suggesting an alternative etiology which remains under research [6,14,15,16]. A possible important factor that can induce GCP in unoperated stomachs is chronic atrophic gastritis [14]. In the present case, the patient’s history was negative for previous gastric surgery. Matsushita et al. [19] proposed to name GCP found in unoperated stomachs heterotopic submucosal cysts, to indicate a different etiology with respect to operated-on stomachs.

Recently, Du et al. published an updated literature review on this topic. Between 1972 and 2014, a total of 52 cases of GCP have been published. Most of the patients were male, older than 50 years old, and no significant difference was highlighted between the operated and unoperated-on patients, with 52% of them having received previous gastric surgery [20].

Symptoms are unspecific and uncommon. When they occur, patients typically present with abdominal pain, fullness, loss of weight, upper-GI bleeding and anemia. The lack of specific symptoms may contribute to the low rate of GCP diagnosis [21].

In the presented case, the patient was admitted to our department for upper-GI bleeding and anemia. GCP is often underdiagnosed because of unspecific clinical presentation and imaging manifestation; therefore, no standardized algorithm for its diagnostic process has been identified yet. In our experience, the first exam performed was an upper-GI endoscopy to identify the cause of the melena and upper-GI bleeding, which revealed a submucosal broad-based lesion of the antrum. Different authors agree on the endoscopic features of GCP, which usually manifests as a submucosal tumor-like lesion or polypoid or similar to early gastric cancer [14,22]. Being mostly limited to the mucosa, endoscopic biopsies are usually not diagnostic, as in our experience. Therefore, further examinations are necessary to better characterize the lesion. Our patient underwent a CE-CT to delineate the size of the lesion, its deep gastric wall involvement, and its lymph-node status. At CE-CT, GCP lesions can appear as spheric or hemispheric, often with a pedicle—usually with a diameter bigger than 1 cm—and with a heterogeneous cystic content [23]. However, CT is not sufficient to reach a proper diagnosis of GCP, due to the low amount of evidence and to the variety of radiological features present.

As suggested by Okada et al. [21], EUS is indicated in cases of suspected GCP. Recently, increasing interest in the role of EUS for the identification of GCP has been manifested in the literature. Multiple cystic lesions within a thickened submucosal layer or heterogeneous enhancing polypoid lesion with a multiple cystic anechoic space in the submucosa are typical EUS findings of GCP. Moreover, fine needle aspiration and FNB can be performed within EUS, contributing to the diagnostic workup [24,25]. In the present case, EUS and FNB were performed, showing a submucosal antral mass with a heterogeneous echo penetrating the muscolaris propriae. When contrast was administered, the lesion presented hyperechoic septa and anechoic holes, features indicative of GIST [26]. We did not include GCP among the possible differential diagnoses of the lesion, because of its very low incidence and unspecific presentation. However, GCP should always be considered among the possible submucosal gastric lesions, and anechoic lacunae within the lesion found during EUS support its diagnosis.

Regardless of the vague indicative findings, the preoperative diagnosis of GCP is challenging and histological examination is necessary to provide a definitive diagnosis. Therefore, pathological examination of the resected specimen is the gold standard for the diagnosis of GCP [20,22,27]. At histology, GCP presents elongated gastric foveolae with hyperplasia and dilation of the gastric glands that extend to the submucosal layer. The glands usually present with a normal morphology, and this is useful in differential diagnosis with cancerous lesions [22]. At the histological diagnosis, GCP may also be associated with early gastric cancer (EGC), and when this association is found, endoscopic resection can be considered curative [28]. Additionally, multifocal GCP may occur in different locations of the stomach as multiple gastric cystic lesions with the same histopathological features [29].

Different resection techniques have been proposed for GCP—both endoscopic procedures and surgery. Endoscopic Mucosal Resection (EMR) and Endoscopic Submucosal Dissection (ESD) are the preferred methods to treat GCP if the patient is asymptomatic and the dimension of the gastric lesion is compatible with endoscopic resection [14,25,28,30]. Concomitantly, surgical resection has been described in the literature as a validated treatment option for GCP. The surgical approach is validated and necessary when endoscopic treatment is not possible or in symptomatic patients [16]. In this case, the patient was recommended for surgical resection with a preoperative potential diagnosis of GIST. In our center, we usually approach GIST with laparoscopic endoscopic cooperative surgery (LECS)—an innovative technique that minimizes invasiveness, preserving the radicality of the resection [31]. LECS was not feasible in this case, due to the mass’ dimension and location and due to the patient having had previous laparotomic abdominal surgery for trauma—hence, open D0 distal gastric resection was performed. Up to 2012, no cases of local recurrence or distant metastases after surgical resection were reported [32]. In 2014, Wang et al. [33] described a case of GCP recurrence within 6 months after surgical resection. Recurrence rates appear to not be influenced by the treatment strategy, even when multiple cystic lesions are identified [34]. Even though robust evidence is lacking due to the rarity of the disease, a regular endoscopic follow-up is suggested in these patients.

The differential diagnosis of submucosal gastric lesions should include leiomyoma, lipoma, gastrointestinal stromal tumor, lymphoma, leiomyosarcoma, neuroendocrine cell tumors, schwannomas, Menetrier’s disease and GCP [7,21,28,32]. Moreover, when facing a submucosal lesion and suspecting GCP, it is paramount to differentiate GCP from adenocarcinoma of the stomach. Although gastric adenocarcinoma usually begins as a mucosal lesion, gastric cancer can also present with normal mucosa in 20–30% of cases, making GCP diagnosis more challenging [35]. At first, GCP was not considered a malignant lesion. Nevertheless, some authors have suggested that GCP is a possible precancerous lesion at a gastroenterostomy site [4,12,30]. Histological findings in previously operated-on stomachs include GCP in association with atrophic gastritis, intestinal metaplasia and dysplastic changes of the epithelium. Moon et al. described dysplastic changes within GCP in selected cases, suggesting a possible role of the latter in the pathway leading to adenocarcinoma [36]. Matsumoto et al. reported a case of outlet obstruction caused by an adenocarcinoma arising from gastritis cystica profunda [9]. As suggested by Mitomi et al., enhanced expression of Ki-67 and p53 in GCP—indicative of increasing cell proliferation and DNA repair activity—may have a role in the development of adenocarcinoma [37]. Choi et al. retrospectively analyzed 10,728 patients who underwent gastric surgery for adenocarcinoma in their department [38]. GCP was present in the gastric specimen in 161 patients (1,5%)—more frequently in multiple gastric cancer—and it was associated with earlier tumor stages in terms of depth of invasion, lymph node metastasis, lymphatic and perineural involvement. Despite these findings, the overall 5-year survival was not different between patients with GCP and those without GCP. Furthermore, EBV infection of gastric epithelium was higher in patients with gastric cancer associated with GCP, suggesting a possible correlation between these two factors that contributes to carcinoma development. In 2010, Roepke et al. demonstrated that the target deletion of a potassium channel protein (KCNE2) in gastric parietal cells of mice causes preneoplastic phenotype, including GCP [39]. Moreover, KCNE2 expression was observed to be reduced in human gastric cancer and to normally be expressed in non-pathological gastric mucosa, showing a possible role of this protein in proliferative gastric disorders. Recently, Kuwahara et al. demonstrated a loss of KCNE2 expression in both GCP and gastric carcinoma in a surgical specimen. Despite its unknown mechanisms, evidence suggests that the selective loss of KCNE2 expression is correlated with the development of GCP in the human stomach, and subsequent occurrences of cancer [40]. Regardless of the studies presented in this paper, no robust data are available on this subject, and whether or not GCP has a role in gastric malignancies remains controversial.

## 4. Conclusions

In conclusion, GCP is a rare submucosal lesion of the stomach whose preoperative diagnosis may be challenging due to its unspecific symptoms and presentation.

It is important to consider GCP in the differential diagnosis of gastric submucosal lesions—not only when the lesion occurs at anastomotic site of operated stomach, but also in case of gastric submucosal lesions in patients without history of gastric surgery.

No standardized diagnostic algorithm is available in the literature, but EUS may contribute to its identification, detecting anechoic lacunae within the lesion.

With histopathological examination being the gold standard for GCP diagnosis, endoscopic or surgical resection are indicated. An adequate endoscopic follow-up after total excision is advisable, considering the risk of GCP local recurrence.

To date, no consensus has yet been reached on the management strategy for GCP, and further studies are needed to clarify pathological mechanisms of GCP and its role as a precancerous lesion in the development of gastric adenocarcinoma.

Here, we reported a case of GCP that was preoperatively misdiagnosed and then successfully identified at histology in a patient without previous gastric surgery. To our knowledge, this is one of the few cases of GCP in the Western population. Finally, we conducted a review of the literature to highlight GCP’s etiological and clinical features, aiming at increasing scientific knowledge on the topic.

## Figures and Tables

**Figure 1 medicina-59-01770-f001:**
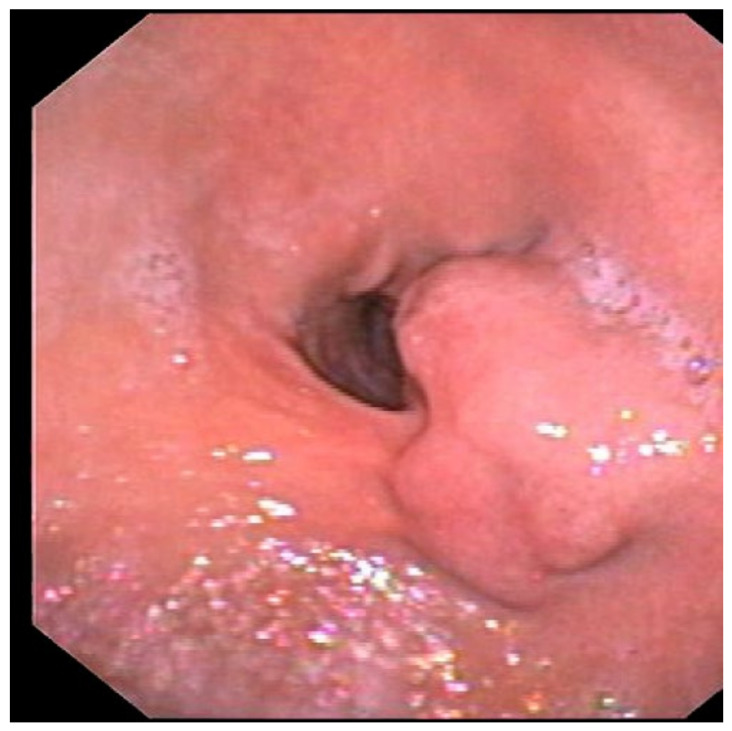
Upper GI endoscopy showing a 25 mm submucosal lesion at the posterior wall of the antrum covered with normal mucosa.

**Figure 2 medicina-59-01770-f002:**
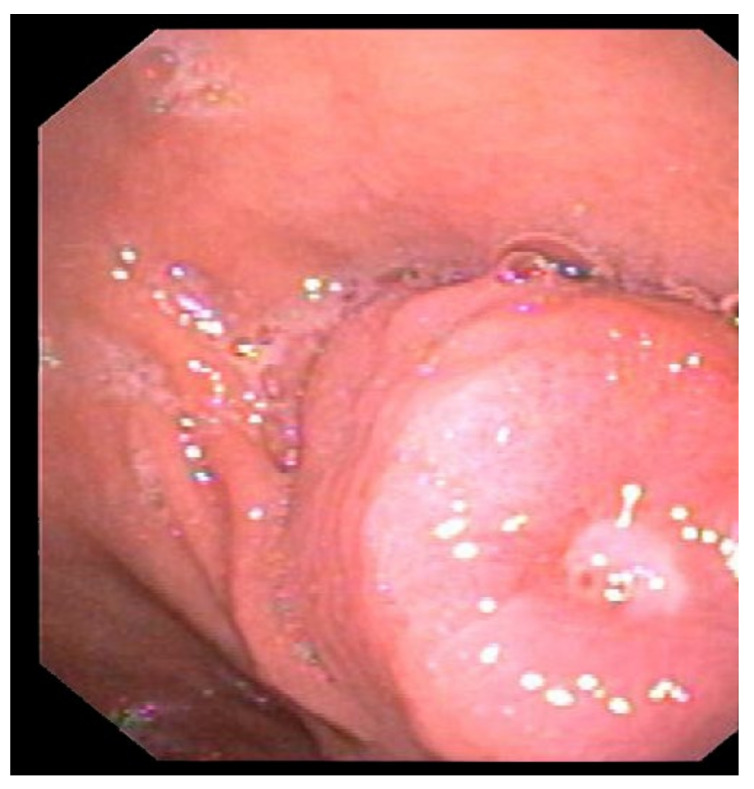
Upper GI endoscopy showing ulceration of the submucosal antral lesion without signs of active bleeding.

**Figure 3 medicina-59-01770-f003:**
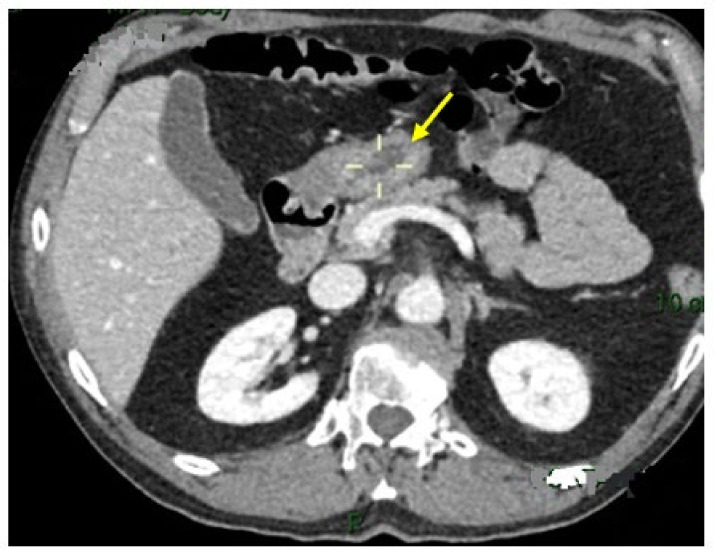
Contrast enhanced computed tomography (CE-CT) showing a 5 × 2 cm lesion of the antrum, with irregular margins. The yellow arrow indicates the lesion evident in the late arterial phase of the CE-CT.

**Figure 4 medicina-59-01770-f004:**
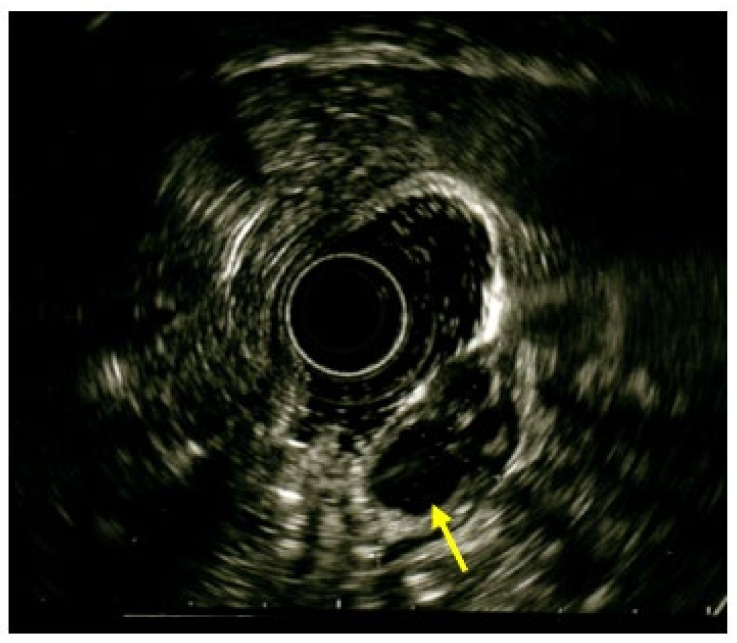
Endoscopic ultrasound showing submucosal heterogeneous echoic mass with multiloculated cystic spaces (yellow arrow).

**Figure 5 medicina-59-01770-f005:**
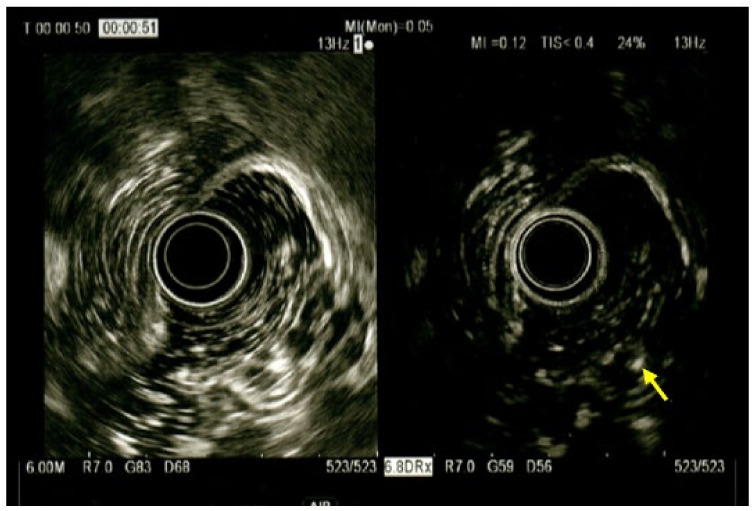
Endoscopic ultrasound after Sonovue^TM^ administration, only the septa of the multiloculated cystic lesion were hyperechoic (yellow arrow) while the holes remained anechoic.

**Figure 6 medicina-59-01770-f006:**
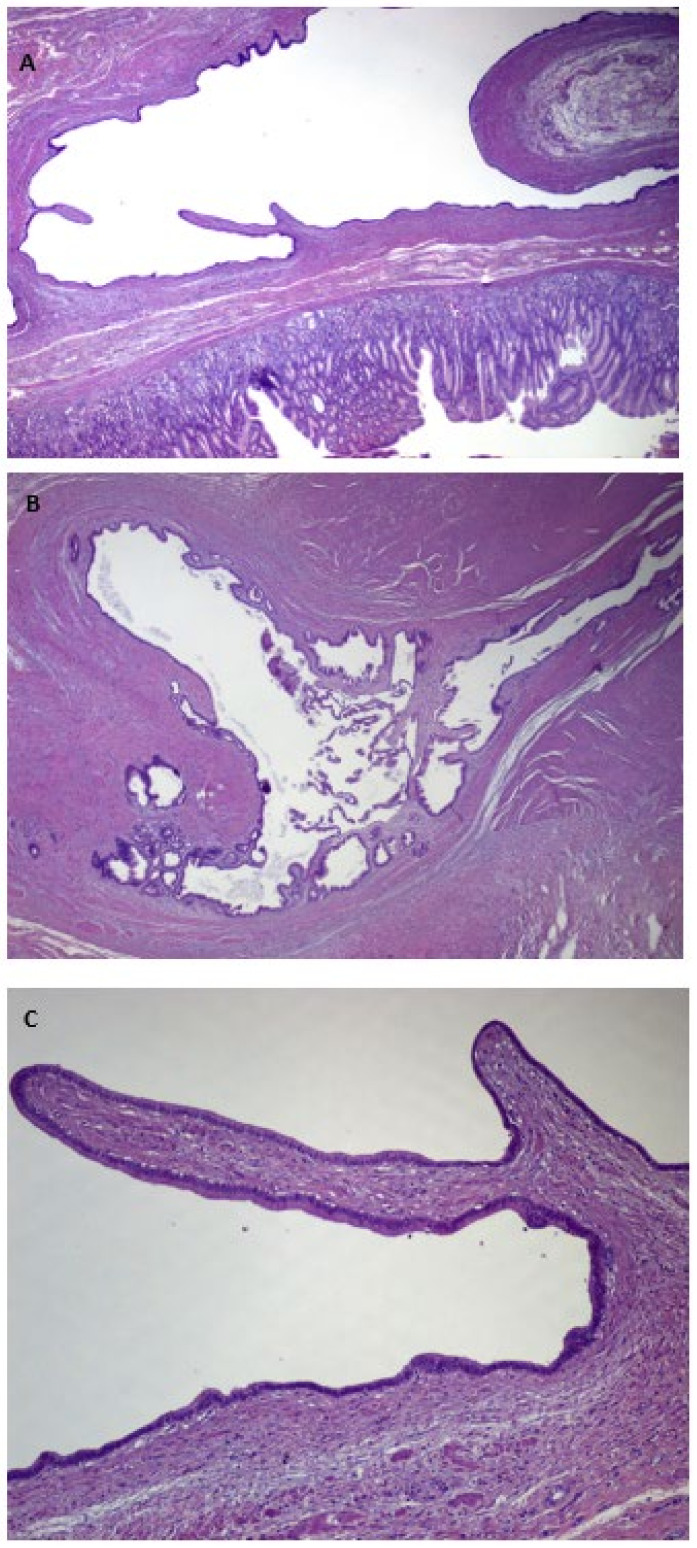
(**A**–**C**) Hematoxylin and eosin staining of the specimen, showing many cystically dilated glands in the submucosa and focally in the muscolaris propriae.

**Figure 7 medicina-59-01770-f007:**
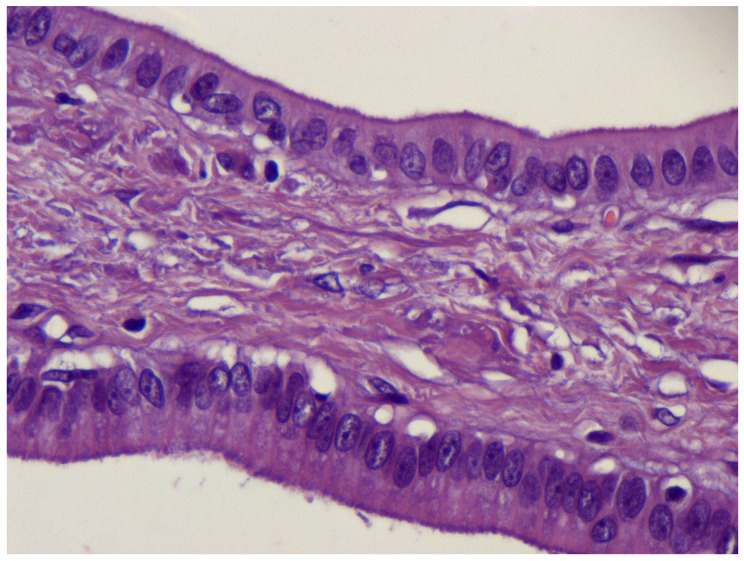
Microscopic examination with hematoxylin and eosin staining, showing chronic inflammation and eosinophilic infiltration in the mucosa.

## Data Availability

No new data were created or analyzed in this study. Data sharing is not applicable to this article.

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
