# Peer review of "Gastritis Cystica Profunda: A Rare Disease, a Challenging Diagnosis, and an Uncertain Malignant Potential: A Case Report and Review of the Literature"

_medicina, 2023, doi:10.3390/medicina59101770_

Round 1

Reviewer 1 Report

In this case report, the authors presented a 61 years old patient with a benign lesion of Gastritis Cystica Profunda that benefit of distal gastrectomy with Billroth II reconstruction. The pathology seems not so uncommon according to the cited literature (Choi et al.) from the discussion section; but, for the presented case, the lesion had a particular localization: on the posterior wall of the gastric antrum, while the majority of cases are located in the fundus area, according the existent evidences. I appreciate that the case is nicely documented, with representative figures, but this article is penurious regarding the elements of novelty over the pathology, diagnostic and treatment.

I truly appreciate the efforts of the authors to write this article, and I can suggest some revisions below:

I appreciate the nice introduction into the subject, starting from the first case officially reported in 1947.

In the line 50 please see if “unclear” or “not yet elucidated” is not better in the sentence “suggesting another not clear pathogenetic mechanism.”

In line 63 please see if “without previous surgery/intervention” is not better instead of “unoperated stomach”

In line 79 where you mentioned “anticoagulant oral therapy”, maybe you can mention the Common international name of the active compound and the dosage/posology into the brackets to be more clear for readers.

Also, in line 80, where you mentioned blood transfusion, maybe you can mention also the blood product that the patient received and how many units. If you had another hemoleucogram maybe you can mention also how much the hemoglobin level (Hb) increased from 8.8g/dl after transfusion. Please pay also attention to numbers you use. In English, you should use “8.8” instead of “8,8”. Please check this for the entire article. The units of measurement for Hb level is “g/dl” (Billett, H.H. Hemoglobin and Hematocrit. In Clinical Methods: The History, Physical, and Laboratory Examinations; Walker, H.K., Hall, W.D., Hurst, J.W., Eds.; Butterworths: Boston, 1990 ISBN 978-0-409-90077-4.).

Maybe some intraoperative pictures of the lesion are also available. Those will complete nicely the images from endoscopy (where the exposure is not very clear), for a better understanding of the case.

I can also suggest if is not very difficult to use some arrows or marks to indicate the lesions and the cysts on the Figures, with a brief description for the unexperienced readers. On figure 3, line 97 (the CT with contrast) you have a mark that is not explained below the figure. Please explain any particularities below. Also you can mention the phase for the CT examination. It seems like a late-arterial phase, but you need to check this information with the radiologist that performed the investigation (https://radiopaedia.org/articles/contrast-phases).

Maybe you can bring some more details about the reintervention: symptoms at presentation, the place of the incarceration, do we have a defective technique during the first surgery that can precipitate a reintervention? Is that related to the pathology, to patient status and activity or to the technique? Some pictures of the lesions from the reintervention are recommended. Do we expect to have reintervention for this pathology?

In the line 129 please pay attention you used small letters (a,b,c) while on the picture you have capitals (A, B, C). Also, not all the letters are rightly positioned on the upper left corner, some are outside the Figure. Please try to use the same manner for all of them. Also please mention the particularities if you have for A, B and C.

On Figure 7 (line 138) you have a different zooming (x). It would be nice to present the zooming also for the pathology images, below the Figure. The coloration and technique are also recommended to be mentioned, especially because the pathology result is crucial for diagnostic.

In the line 176 where you presented the symptoms maybe you can use some citations.

Please see again the phrase from line 277, from Conclusion section, maybe some words are missing.

Thank you and good luck further!

Small revision.

Author Response

Response to Reviewer 1

Thank you for giving us the opportunity to revise our manuscript. The reviews highlighted a variety of ways in which our paper could be implemented. We have worked in this round to develop an adjusted version of our manuscript that responds to the concerns that you and the reviewers raised. We hope that you will find the current draft more valid for publication.

Comments to the authors

In this case report, the authors presented a 61 years old patient with a benign lesion of Gastritis Cystica Profunda that benefit of distal gastrectomy with Billroth II reconstruction. The pathology seems not so uncommon according to the cited literature (Choi et al.) from the discussion section; but, for the presented case, the lesion had a particular localization: on the posterior wall of the gastric antrum, while the majority of cases are located in the fundus area, according the existent evidences. I appreciate that the case is nicely documented, with representative figures, but this article is penurious regarding the elements of novelty over the pathology, diagnostic and treatment.

 Thank you again for your comments. Gastritis cystica profunda is still considered an uncommon finding and the literature is still poor on the topic. This stands true especially regarding the Western population, in which only few case reports have been published. Therefore, we believe that our manuscript may contribute to increase the awareness in this pathology in Western countries. 

I truly appreciate the efforts of the authors to write this article, and I can suggest some revisions below.

I appreciate the nice introduction into the subject, starting from the first case officially reported in 1947.

We really appreciate your comments. We accurately revised the clinical records of the patient as well as the literature on the topic in order to present updated information.

In the line 50 please see if “unclear” or “not yet elucidated” is not better in the sentence “suggesting another not clear pathogenetic mechanism.”

In line 63 please see if “without previous surgery/intervention” is not better instead of “unoperated stomach”

 We modified the text according to your language suggestions.

In line 79 where you mentioned “anticoagulant oral therapy”, maybe you can mention the Common international name of the active compound and the dosage/posology into the brackets to be more clear for readers.

Also, in line 80, where you mentioned blood transfusion, maybe you can mention also the blood product that the patient received and how many units. If you had another hemoleucogram maybe you can mention also how much the hemoglobin level (Hb) increased from 8.8g/dl after transfusion. Please pay also attention to numbers you use. In English, you should use “8.8” instead of “8,8”. Please check this for the entire article. The units of measurement for Hb level is “g/dl” (Billett, H.H. Hemoglobin and Hematocrit. In Clinical Methods: The History, Physical, and Laboratory Examinations; Walker, H.K., Hall, W.D., Hurst, J.W., Eds.; Butterworths: Boston, 1990 ISBN 978-0-409-90077-4.).

 We adjusted the lacking information in the manuscript according to your suggestions. We also implemented details on transfusions and therapy.

Maybe some intraoperative pictures of the lesion are also available. Those will complete nicely the images from endoscopy (where the exposure is not very clear), for a better understanding of the case.

Unfortunately, no intraoperative pictures were taken, especially because the peculiar histological diagnosis was unexpected.

I can also suggest if is not very difficult to use some arrows or marks to indicate the lesions and the cysts on the Figures, with a brief description for the unexperienced readers. On figure 3, line 97 (the CT with contrast) you have a mark that is not explained below the figure. Please explain any particularities below. Also you can mention the phase for the CT examination. It seems like a late-arterial phase, but you need to check this information with the radiologist that performed the investigation (https://radiopaedia.org/articles/contrast-phases).

 Thank you for your suggestions. We used arrows to highlight relevant details of the figures and we specified the CT phase.

Maybe you can bring some more details about the reintervention: symptoms at presentation, the place of the incarceration, do we have a defective technique during the first surgery that can precipitate a reintervention? Is that related to the pathology, to patient status and activity or to the technique? Some pictures of the lesions from the reintervention are recommended. Do we expect to have reintervention for this pathology?

Thank you for this comment. Our manuscript was lacking this information. We revised the clinical records in order to report details of the reintervention. Unfortunately, no pictures from the operating room are available. The bowel obstruction consequent to postoperative adhesions was not related to the technique used in the first surgery. However, we are aware that open surgery increases the risk of adhesions with respect to minimally invasive approaches. To our knowledge, this pathology does not increase the risk of reintervention.

In the line 129 please pay attention you used small letters (a,b,c) while on the picture you have capitals (A, B, C). Also, not all the letters are rightly positioned on the upper left corner, some are outside the Figure. Please try to use the same manner for all of them. Also please mention the particularities if you have for A, B and C. 

On Figure 7 (line 138) you have a different zooming (x). It would be nice to present the zooming also for the pathology images, below the Figure. The coloration and technique are also recommended to be mentioned, especially because the pathology result is crucial for diagnostic.

We implemented these figures and followed your suggestions.  

In the line 176 where you presented the symptoms maybe you can use some citations.

We inserted the citation [23].

Please see again the phrase from line 277, from Conclusion section, maybe some words are missing.

Thank you for your advice. As you suggested, the phrase was not fluent and we revised it.

Reviewer 2 Report

First of all, I appreciate the opportunity to review this interesting article. This is an article presenting a clinical case and review, in which the diagnostic process and presentation of a patient diagnosed with "Deep Cystic Gastritis" is described. The article adequately presents an introduction to this rare disease, as well as its diagnostic approach. Since the definitive diagnosis of this pathology depends exclusively on the histopathological study, the initial approach hardly takes it into account as a possibility.

In general, I find the article well written and of great clinical use. I just have some minor observations/recommendations:

1.- In the description of the clinical history, it would be advisable to add the medications you are currently taking, including doses, as well as specify the name and dose of the oral anticoagulant treatment prior to hospital admission. It would also be worth mentioning if you have a history of episodes similar to this one, and the length of time you had the clinical manifestations. Also mention history of smoking and alcohol consumption, if available.

2.- Mention a history of gastrointestinal signs and symptoms, especially if the patient had a previous diagnosis of gastritis, gastroesophageal reflux or others, as well as if they had a history of previous diagnosis and treatment for H. pylori.

3.- Mention the follow-up time after definitive hospital discharge.

4.- In the scientific literature, is there any reference that mentions the frequency of presentation of the lesion according to the gastrointestinal anatomical area? (if so, please enter this information)

5.- Is there any information on its prevalence and incidence, whether local or global?

I believe that by answering these doubts and complementing them with the available information, this article could be published and thus add important information about this rare, little-studied gastrointestinal entity.

Author Response

Response to Reviewer 2

Thank you for giving us the opportunity to revise our manuscript. The reviews highlighted a variety of ways in which our paper could be implemented. We have worked in this round to develop an adjusted version of our manuscript that responds to the concerns that you and the reviewers raised. We hope that you will find the current draft more valid for publication.

Comments to the authors

First of all, I appreciate the opportunity to review this interesting article. This is an article presenting a clinical case and review, in which the diagnostic process and presentation of a patient diagnosed with "Deep Cystic Gastritis" is described. The article adequately presents an introduction to this rare disease, as well as its diagnostic approach. Since the definitive diagnosis of this pathology depends exclusively on the histopathological study, the initial approach hardly takes it into account as a possibility. In general, I find the article well written and of great clinical use. I just have some minor observations/recommendations.

Thank you again for your kind comments. Gastritis cystica profunda is still considered an uncommon finding and the literature is still poor, therefore it is often underdiagnosed. With our manuscript we would like to increase awareness on the topic.

1.- In the description of the clinical history, it would be advisable to add the medications you are currently taking, including doses, as well as specify the name and dose of the oral anticoagulant treatment prior to hospital admission. It would also be worth mentioning if you have a history of episodes similar to this one, and the length of time you had the clinical manifestations. Also mention history of smoking and alcohol consumption, if available.

We adjusted the lacking information in the manuscript according to your suggestions. Unfortunately, no information are available on history of smoking and alcohol consumption.

2.- Mention a history of gastrointestinal signs and symptoms, especially if the patient had a previous diagnosis of gastritis, gastroesophageal reflux or others, as well as if they had a history of previous diagnosis and treatment for H. pylori.

We added this information in the manuscript. The medical history of the patient was negative for previous gastrointestinal disorders.

3.- Mention the follow-up time after definitive hospital discharge.

Information has been added in the text. At present, the follow-up is still going on.

4.- In the scientific literature, is there any reference that mentions the frequency of presentation of the lesion according to the gastrointestinal anatomical area? (if so, please enter this information)

Data on the frequency of presentation according to the anatomical can be found in ref. [15,16].

5.- Is there any information on its prevalence and incidence, whether local or global?

Information on this aspect is not available in the literature, due to the poor number of cases reported.

Reviewer 3 Report

A well-written case report with adequate review of literature 

English is largely accpetable

Author Response

Dear Reviewer, 

Thank you for the time spent reading and reviewing our paper and thank you for your comments. 

My team and I appreciate your interest in our paper and the positive opinion you expressed.

As you suggested, we improved the language.